# A Novel Green Micellar HPLC-UV Method for the Estimation of Vandetanib in Pure Form, Human Urine, Human Plasma and Human Liver Microsomes Matrices with Application to Metabolic Stability Evaluation

**DOI:** 10.3390/molecules27249038

**Published:** 2022-12-18

**Authors:** Mohammed M. Alanazi, Ahmad J. Obaidullah, Mohamed W. Attwa

**Affiliations:** 1Department of Pharmaceutical Chemistry, College of Pharmacy, King Saud University, P.O. Box 2457, Riyadh 11451, Saudi Arabia; 2Students’ University Hospital, Mansoura University, Mansoura 35516, Egypt

**Keywords:** green chemistry, vandetanib, human urine, human plasma, human liver microsomes

## Abstract

Vandetanib (Caprelsa^®^; VNB) is a prescription medicine that is used for the treatment of medullary thyroid cancer that has disrupted other body parts or that cannot be removed by surgery. It is considered a tyrosine kinase inhibitor (TKI). Fast, sensitive and validated HPLC–UV was established for VNB quantification in pure human biological fluids (urine and plasma) and human liver microsomes (HLMs). This analytical methodology was applied also to the metabolic stability assessment of VNB. This method was performed using a phenyl column (250 mm × 4.6 mm id, 5 µm particle size). A sodium dodecyl sulphate solution (0.05 M, pH 3.0 using 0.02 M orthophosphoric acid) containing 0.3% triethylamine and 10% n-butanol was used as a mobile phase and was pumped isocratically at a flow rate of 0.7 mL/min and at a 260 nm detection wavelength. The total elution time was 6 min with an injection volume of 20 μL. The linearity of the established methodology ranged from 30 to 500 ng/mL in pure form and 50 to 500 ng/mL (r^2^ ≥ 0.9994) in human biological fluids and HLMs. No significant interference from the matrix components was observed. The proposed methodology revealed the benefits of being green, reliable and economic.

## 1. Introduction

Vandetanib (Caprelsa^®^, Figure 1), or *N*-(4-bromo-2-fluorophenyl)-6-methoxy-7-((1-methylpiperidin-4yl) methoxy) quinazolin-4-amine [1,2], was developed by AstraZeneca, who later sold the rights to Sanofi in 2015, and is a unique tyrosine kinase inhibitors (TKIs) used in the treatment of aggressive and symptomatic medullary carcinoma in thyroid glands. It acts through the inhibition of various receptors of human cells, such as the epidermal growth factor receptor, RET-tyrosine kinase and others [3,4,5]. Additionally, VNB can be utilized either alone or administered with other anti-cancer drugs for routine treatment, such as radiotherapy or chemotherapy for handling other types of tumors [6,7].

There are a limited number of reported methods for the quantification of VNB, such as spectrofluorimetry and LC/MS [8,9]. Both published analytical methodologies were used to estimate VNB in rat liver microsomes, plasma and urine using fluorescence detectors or LC-MS/MS detectors. Additionally, VNB was estimated either alone or in combination with other drugs [10,11,12,13,14]. The micellar liquid chromatographic analytical method is one mode of eco-friendly analytical chemistry and is reported for the separation of a number of drugs [15,16,17]. Pharmacokinetic studies on VNB in humans were performed in phase I dose escalation studies [18] and in phase II trials at VNB doses of 100 and 300 mg/day [19]. The outcomes of these reported studies revealed that VNB is widely distributed, slowly absorbed and slowly excreted with a half-life (t_1/2_) of 120 h [20].

The analytical chemistry field has been seeking to develop time-saving, cost-effective and green methods [21]. The developed method focused on reducing the use of organic solvents and utilizing more eco-friendly ones that separate pharmaceutical residues in biological fluids quickly and with high efficiency and resolution. This is a challenge in official control analysis and can be achieved by using surfactants, such as sodium dodecyl sulphate (SDS). SDS, with its high solubilizing effect on protein, enables the direct injection of biological fluids into HPLC columns, thus releasing drugs from protein-bound sites and prohibiting interferences with drug estimation, thus extending the lifetime of the column [22].

Since plasma and urine are complex matrices, diluting them with a micellar mobile phase and filtration before the injection of the biological samples decreases the width of the protein band in the chromatograms, thus removing the interferences with drug estimation. The lifetime of the column is extended by the dilution of samples [22]. About 3.4% of an adult dose of VNB (300 mg per day) is excreted unmetabolized in the urine [11], so the target of the current research was to develop a green analytical methodology for the quantification of VNB in human biological fluids (urine and plasma). Searching the literature revealed that there were no green HPLC-UV method was published for the quantification of VNB in HLM matrices or for the assessment of VNB metabolic stability.

Therefore, these outcomes encouraged us to establish a validated and efficient method for the quantification of VNB levels with high accuracy and precision. Accordingly, an HPLC-UV analytical methodology was used for the quantification of VNB conc. in different matrices, including human urine, human plasma and HLMs. The proposed methodology was applied for assessing the metabolic stability of VNB using the rate of VNB’s disappearance during incubation with HLMs. When a test compound was slowly metabolized, that resulted in an accumulation of the drug inside the human body, leading to side effects after multiple doses [23,24,25,26].

## 2. Results

### 2.1. Chromatographic Separation

Different chromatographic conditions were adjusted after many trials to obtain the best resolution, highest sensitivity and a symmetrical peak shape. Different percentages of 0.05 M SDS and *n*-butanol were tested. An isocratic elution using a mixture of 0.05 M sodium dodecyl sulfate (SDS) solution containing 10% *n*-butanol and 0.3% triethylamine and a pH adjusted to 3 using 0.02 M phosphoric acid showed excellent peak shape and the best chromatographic resolutions from endogenous matrix component peaks. The elution was achieved in 4 min, and the run time was 6 min (Figure 2).

### 2.2. Analytical Characteristics and Method Validation

#### 2.2.1. Linearity

The linearity was assessed via analysis sequence of different conc. of each analyte. The desired conc. ranges and the parameters for regression equations are given in Table 1.

#### 2.2.2. Detection (LOD) and Quantification (LOQ) Limits

The LOD and LOQ were calculated as LOQ = 10 (SD_i_/S) and LOD = 3.3 (SD_i_/S), where ‘S’ is the slope of the calibration graph and ‘SD_i_’ is the standard deviation of the intercept, Table 2.

#### 2.2.3. Suitability of the System

System suitability was accomplished during the method’s optimization. This test was conducted by injecting the standard mixture of the three replicates, and the validation parameters were computed as reported by the USP [27]. The different system suitability parameters were abridged in Table 2.

#### 2.2.4. Precision and Accuracy

Intra- and inter-day accuracy and precision of the three quality control (low, medium and high) level conc. were reproducible, as found in Table 3.

#### 2.2.5. Stability

Stability experiments were finished under various conditions using the three quality control (low, med and high) samples. It was shown that no observed VNB losses occurred while handling the samples under the test conditions (Table 4).

### 2.3. Applications

#### 2.3.1. Applications to CAPRELSA Tablets

The proposed method was applied for the analysis of CAPRELSA tablets. The outcomes obtained matched with the results reported in 8], where no significant variance between the two was noticed, as seen in Table 5. Student’s *t*-test and the variance ratio F-test were applied in statistical data analysis at a 95% confidence level [28].

#### 2.3.2. Metabolic Stability

The metabolic reactions of VNB in the HLMs were stopped at certain time intervals. The curve for the metabolic stability of VNB in the HLMs was determined by plotting the % remaining of the VNB conc. (compared to the zero-time point conc.) against incubation time (Figure 3). In vitro half-life was calculated from the linear part regression equation of the constructed curve.

In vitro t ½ = 0.693/Slope;

The slope equals 0.01;

In vitro t ½ = 0.693/0.01 = 63.01 min.

Accordingly, the intensity clearance was calculated according to the in vitro t ½ method [20] as in the following formula:CL_int_ = (0.693/In vitro t ½) ∗ (µL incubation/mg HLMs)
CL_int_ = (0.693/63.01) ∗ (1000/1) = 11.0 µL/min/mg

## 3. Discussion

The analytical HPLC-UV methodology showed a high capability of the VNB and IS estimation with good resolution within a short run time (6 min) to reduce the amount of the consumed drug. The use of micellar solutions on HLMs and human plasma matrices eliminates the need for extraction procedures. The validation studies, according to the FDA and ICH, showed the ability of the proposed analytical methodology to control the official analysis of pure VNB and the quantification of VNB in real biological fluids (plasma and urine), as well as its application in assessing the metabolic stability of VNB in HLMs. The suggested methodology showed the benefits of being green, simple, reliable and cost-effective. The proposed method was applied to estimate amounts of VNB from CAPRELSA tablets. Table 5 shows that there is no difference in the results of the Student’s *t*-test and the variance ratio F-test. The intrinsic capacity of VNB metabolism in HLMs (CL_int_ = 11.0 µL/min/mg) was low with a long in vitro t ½ (approximately 63.01 min), revealing that VNB is slowly excreted by the liver from the blood. Thus, VNB is considered a low-extraction-ratio drug, and this slow extraction can lead to side effects that have been reported in the literature [11,29]. Therefore, the VNB level in human plasma and urine should be monitored using the developed HPLC-UV analytical method to avoid drug accumulation leading to side effects, as was proposed in the metabolic stability study.

## 4. Experimental Section

### 4.1. Reagents and Chemicals

Vandetanib (VNB) was bought from LC Laboratories (Woburn, MA, USA). Acetonitrile (HPLC grade) was bought from Sigma-Aldrich (West Chester, PA, USA). A Milli-Q plus purification instrument from Millipore (Waters, Bedford, MA, USA) was utilized to generate HPLC-grade water. Pooled human liver microsomes (HLMs; (M0567) from male donors (20 mg/mL in 250 mM sucrose buffer) were purchased from Sigma-Aldrich (St. Louis, MO, USA) and stored at −70 °C until use. Caprelsa (vandetanib)^®^ (300 mg) tablets were procured from a local market. Phosphate buffer 0.05 M, sodium dodecyl sulphate (SDS), orthophosphoric acid and triethylamine (TEA) were procured from El-Nasr Pharmaceutical Chemicals (Cairo, Egypt). N-butanol (HPLC-grade) was purchased from Sigma–Aldrich (St. Louis, MO, USA). Plasma and urine samples were gathered from healthy volunteers (*n* = 3) and were kept at −20 °C until use after gentle thawing. All HPLC-grade solvents and deionized water were used in this work. The study design used in vitro experiments with human liver microsomes commercially available from Sigma-Aldrich, thus making it exempts from the need for ethical approval. Plasma and urine from healthy volunteers were supplied by a local hospital after the consent of volunteers was received, and the study was conducted according to the guidelines of the Declaration of Helsinki.

### 4.2. Apparatus

The HPLC system consists of a Perkin Elmer^TM^ Series 200 Chromatograph with a 20 µL loop and UV/VIS detector, supplied with an injector valve of Rheodyne. The analytical column is a Nucleosil 100-5 Phenyl column (250 mm length × 4.6 mm internal diameter, 5 µm particle size), Macherey–Nagel, and a USA.0.45μm membrane filter (Millipore, Ireland) was used for mobile phase filtration. The pH meter was purchased from Belgium and was model NV P-901. An automatic tablet dissolution tester (Abbott Laboratories Corp., Princeton, NJ, USA) was used for the in vitro dissolution test.

### 4.3. Chromatographic Conditions

RP-HPLC assays were carried out isocratically with a flow rate of 0.7 mL/min at 25 °C. A solution of 0.05 M SDS (pH 3.0) containing 0.3% TEA and 10% *n*-butanol was used as aqueous phase. Orthophosphoric acid (0.02 M) was used for pH adjustment. The wavelength of the detector was set at 260 nm. Further purification of the samples was performed through filtration using 0.45 µm disposable filters before injecting a 20 µL sample. All determinations were performed at 25 °C. The analysis was completed within 6 min.

### 4.4. Preparation of VNB Calibration Levels

VNB stock solution (1 mg/mL) was dissolved in ACN, and further dilution (100-fold) using the mobile phase was performed to prepare working solution 1 (WK1, 10 μg/mL). Dilution (10-fold) with the mobile phase was performed to prepare WK2 (1 μg/mL).

### 4.5. Construction of VNB Calibration Curve

For pure VNB, the calibration levels were prepared by dilution of the WK2 and of the VNB within their mobile phase to reach conc. ranges of 30–500.

For urine: dilution (ten-fold) of the urine sample was performed with water, then transferred to a centrifuge tube. Centrifugation was performed at 1500 rpm for 60 s. to remove the particulate matter and the cell debris. Second: filtration of the supernatant was performed through 0.45 µm syringe filters. Then, the calculated amount of the VNB WK2 was added to 0.5 mL of the filtered urine, then completed to mark with mobile phase in 10 mL volumetric flasks. This was followed by mixing for 60 s to prepare the final conc. ranges of 50–500 ng/mL.

Plasma from healthy volunteers was supplied by a local hospital, after the consent of doctors and volunteers was obtained. Plasma samples were frozen at −20 °C. Plasma samples were then thawed at room temperature before use. Next, 0.5 mL of blank plasma was placed in 10 mL volumetric flasks, then spiked with a different conc. of the standard levels and mixed well for 60 s. Then, the samples were completed to the mark with the mobile phase, generating conc. ranges of 50–500 ng·mL^−1^.

For HLMs, the matrix was spiked with calculated volumes of WK2 to conc. ranges of 50–500 ng·mL^−1^. First, 1 mL of all samples was taken and then 1 mL of 0.1 M buffer (NaOH/glycine, pH 9.5) was added. Shaking for 30 s was performed. The protein extraction methodology was utilized for VNB extraction, as it is the standard extraction method for metabolic stability experiments because ACN is used as a quenching and precipitating agent [30,31,32]. Two milliliters of ACN was centrifuged was performed at 14,000 rpm (12 min at 4 °C) to remove precipitated proteins. Filtration of the supernatants was performed using a 0.22 μm syringe filter. Evaporation of the filtrates was performed under a stream of nitrogen, followed by dissolving the precipitates in 1 mL of the mobile phase. The same steps were repeated to make the blank mobile phase without VNB to verify the absence of any intervention at the elution time of VNB.

Samples were injected three times for each conc. To plot the calibration graph, the final drug conc. was plotted versus the corresponding peak area, and the regression equation was calculated.

### 4.6. Quality Control Sample Preparation

Three conc. (low, medium and high) were chosen as quality control samples of the bulk drug and were prepared in methanol at (40, 200 and 400 ng/mL) and at (60, 200 and 400 ng/mL) for human urine, human plasma and HLMs.

### 4.7. Method Validation

Validation of the analytical methodology was performed following the guidelines for analytical methods and procedures by the FDA [33] and the general instructions of the International Conference on Harmonization (ICH) [34,35].

#### 4.7.1. Linearity

Calibration curves in each medium were constructed to compute the linearity of the supposed analytical methodology.

#### 4.7.2. Detection (LOD) and Quantification Limit (LOQ)

The limit of quantification (LOQ) was computed as LOQ = 10 (SD_i_/S). The limit of detection (LOD) was computed as LOD = 3.3 (SD_i_/S), where S is the slope of the calibration line and SD represents the standard deviation of the intercept of the regression line of the calibration curve.

#### 4.7.3. Precision and Accuracy

The accuracy and precision of the analytical methodology were tested at the three quality control levels (low, medium and high QC conc.) for each drug. Accuracy was evaluated in triplicate at three different conc. levels for each analyte. Five replicates of the calibration curve evaluation were performed in the same day to check intra-day precision, and evaluations were performed on three consecutive days (five replicates in the first day and three replicates in the two following days for inter-day precision).

#### 4.7.4. Stability

To estimate stability of VNB in different matrices (human plasma, human urine and HLMs), evaluation of three repeats of QC samples were performed under different storage conditions. Precision and accuracy values were calculated utilizing regression equation of freshly prepared calibration curves in different matrices. First: VNB QC samples were stored at room temperature (T) for 8 h to evaluate the bench-top stability of VNB. Second: estimation of VNB’s stability in QC samples after freezing at −20 °C and thawing at room T for three cycles was performed to evaluate the VNB stability under freeze–thaw conditions. Third: estimation of VNB stability was attained by quantification of QC samples after their storage at −20 °C for 30 days and after storage at 4 °C for 1 day.

### 4.8. Assay of VNB Tablet Samples

After grinding and weighing five tablets, tablet powder equal to 100 mg VNB was weighed and transported to a 100 mL volumetric flask. Fifty milliliters of ACN was added to the volumetric flask which was then shaken and sonicated (10 min.). Following this, more ACN was used to dilute the volume to 100 mL (until the mark) to prepare the proposed 1 mg/mL conc. Further dilution was performed to achieve the conc. inside the calibration curve.

### 4.9. Metabolic Stability of VNB

The metabolic effect of HLMs on the VNB conc. after incubation with the HLMs matrix (1 mg/mL) was used to assess the metabolic stability of VNB. Incubations of 0.1 μM VNB with HLMs (1 mg/mL) were performed in triplicates. First: incubation for all samples was performed for 10 min to attain 37 °C. Second: initiation of the metabolism pathways was performed through adding 1 mM NADPH (co factor) in phosphate buffer (pH 7.4) containing 3.3 mM MgCl_2_, while the quenching of the metabolic pathway was performed by adding 2 mL of ACN (quenching and precipitating agent) at time points of 0, 2.5, 5, 10, 15, 20, 40, 50, 70, 90 and 120 min. The extraction of VNB was performed as previously discussed. Conc. of VNB in the HLMs matrix were computed from the generated regression equation of the freshly constructed VNB calibration curve.

## 5. Conclusions

A green HPLC-UV methodo using a phenyl column was established in this study. The method showed a high capability for the determination of analytes, with good separation within a short run time (6 min). The proposed eco-friendly methodology minimizes the utilization of organic mobile phases and demonstrates high sensitivity, efficiency and accuracy. Due to the solubilizing effect of micellar solutions on macromolecules, including proteins of the plasma matrix, no purification and extraction procedures were needed before sample injection. This avoids any tedious steps, variables and incomplete recovery. The validation studies according to the FDA and ICH, showed the ability of the proposed analytical methodology to control the official analysis of pure VNB; the quantification of VNB in real biological fluids (plasma and urine); and in assessing the metabolic stability of VNB in HLMs. The low CL_int_ (11.0 µL/min/mg) and the long in vitro t ½ (63.01 min) of VNB in HLMs revealed that VNB is slowly excreted from the blood by the liver, leading to side effects that match those reported in the literature. Therefore, the VNB level in human plasma and urine should be monitored using the developed HPLC-UV analytical method to avoid drug accumulation leading to side effects, as was proposed from the metabolic stability study. No significant interference from the matrix (urine, plasma and HLMs) components was observed. The suggested methodology demonstrated the benefits of being green, simple, reliable and cost-effective. The established HPLC-UV analytical methodology can also be applied for the quantification of VNB in human plasma and human urine, so it could be used in pharmacokinetic and therapeutic drug monitoring studies in humans.

## Figures and Tables

**Figure 1 molecules-27-09038-f001:**
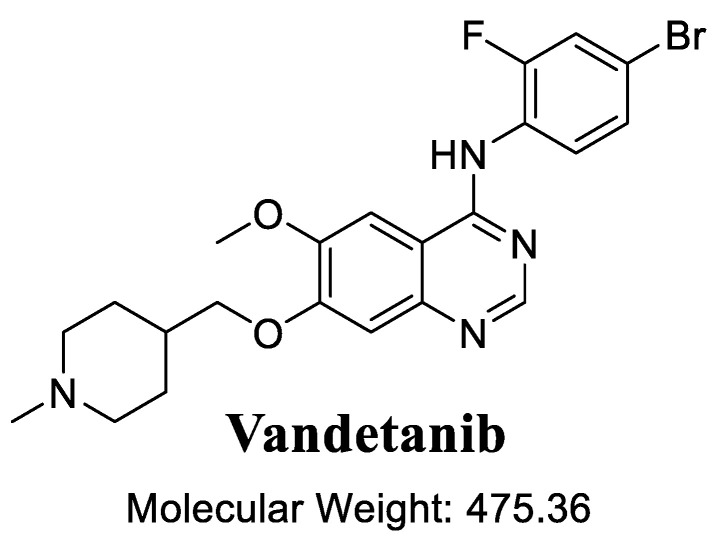
Chemical and molecular structure of vandetanib (VNB).

**Figure 2 molecules-27-09038-f002:**
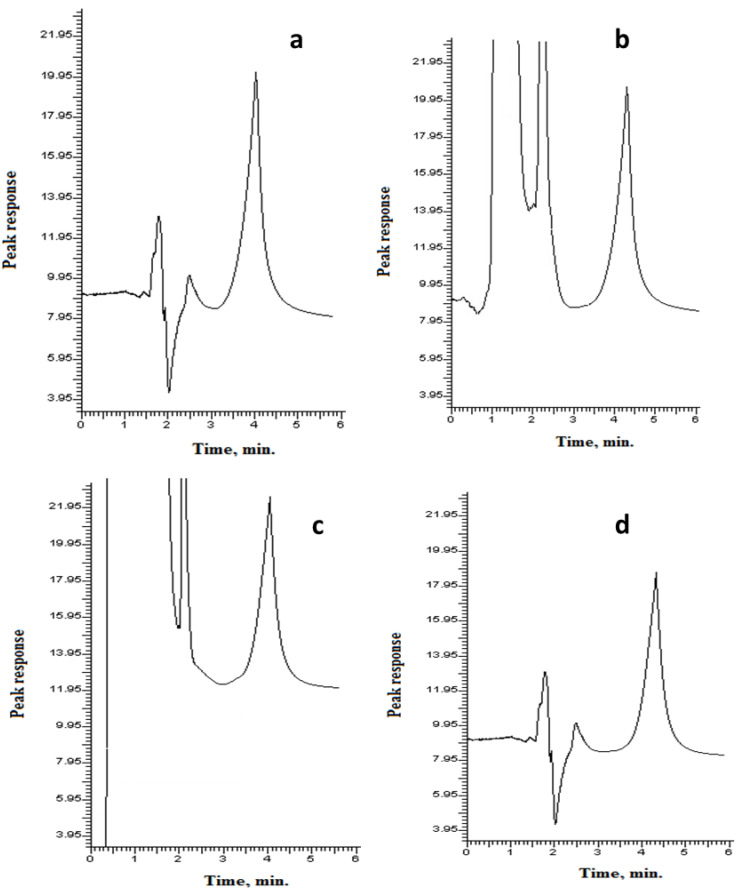
A typical chromatogram for 500 ng/mL of VNB under described chromatographic conditions: 0.3% triethylamine and 5% *n*-butanol in a solution of 0.05 M SDS was adjusted to pH 3 utilizing 0.02 M orthophosphoric acid and pumped at a flow rate of 0.7 mL min^−1^: (**a**) pure, (**b**) human urine, (**c**) human plasma and (**d**) HLMs.

**Figure 3 molecules-27-09038-f003:**
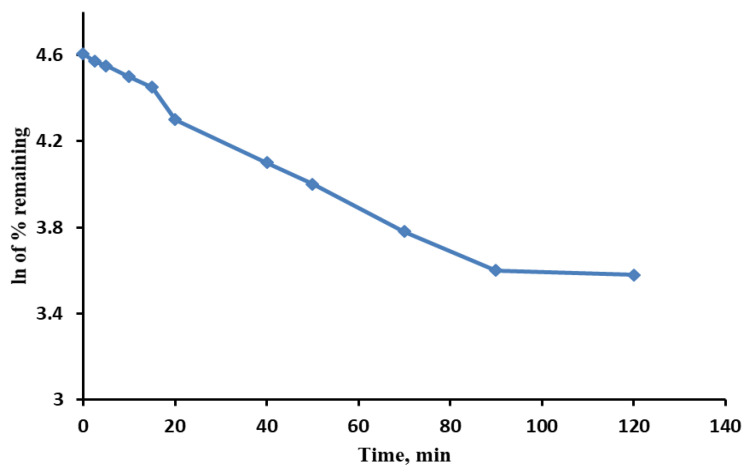
The metabolic stability profile of VNB in HLMs.

**Table 1 molecules-27-09038-t001:** Performance data for the estimation of VNB by the proposed analytical methodology.

Parameter	Pure	Urine	Plasma	HLMs
Linearity range (ng ml^−1^)	(30–500)	(50–500)	(50–500)	(50–500)
Intercept (a)	0.6889	0.6397	0.5748	0.6629
Slope (b)	0.0123	0.0127	0.0128	0.0109
Correlation coefficient (r)	0.9998	0.9995	0.9996	0.9995
S.D. of residuals(S_y/x_)	0.0600	0.0855	0.0808	0.0748
S.D. of intercept (S_a_)	0.0360	0.0561	0.0531	0.0491
S.D. of slope (S_b_)	0.0001	0.0002	0.0002	0.0002
Limit of detection, LOD (ng mL^−1^)	8.8600	13.280	12.450	13.560
Limit of quantitation, LOQ (ng mL^−1^)	29.530	44.270	41.490	45.210

**Table 2 molecules-27-09038-t002:** System suitability test parameters of the established HPLC methodology for the VNB quantification.

Parameter	VNB
No. of theoretical plates, *N*	2025.0
High equivalent theoritical plates, HETP	0.1235
Capacity factor, *k*	1.2500
Tailing factor, T	1.0435
Assymmetry factor, A_f_	1.0870

**Table 3 molecules-27-09038-t003:** Intra-day and inter-day accuracy and precision calculated from quality control (QC) samples for VNB.

**VNB**															
**Inter-Day (*n* = 11)**															
**Conc** **(ng mL^−1^)**	**Pure**			**Conc** **(ng mL^−1^)**	**Urine**			**Conc** **(ng mL^−1^)**	**Plasma**			**Conc** **(ng mL^−1^)**	**HLMs**		
	**Mean ^a^ ± SD**	**Precision**	**%** **Accuracy**		**Mean ^a^ ± SD**	**Precision**	**%** **Accuracy**		**Mean ^a^ ± SD**	**Precision**	**%** **Accuracy**		**Mean ^a^ ± SD**	**Precision**	**%** **Accuracy**
40	39.56 ± 0.72	1.83	98.91	60	59.84 ± 0.81	1.36	99.73	60	59.51 ± 1.02	1.71	99.19	60	59.71 ± 0.68	1.15	99.51
200	199.7 ± 0.44	0.22	99.85	200	199.7 ± 0.78	0.39	99.85	200	199.6 ± 0.44	0.22	99.79	200	199.6 ± 0.66	0.33	99.78
400	399.7 ± 0.46	0.12	99.93	400	399.9 ± 0.84	0.21	99.98	400	399.9 ± 0.94	0.24	99.97	400	399.9 ± 0.58	0.14	99.96
**Intra-Day (*n* = 5)**															
**Conc** **(ng mL^−1^)** **40** **200** **300**	**Pure**			**Conc** **(ng mL^−1^)**	**Urine**			**Conc** **(ng mL^−1^)**	**Plasma**			**Conc** **(ng mL^−1^)**	**HLMs**		
	**Mean ^a^ ± SD**	**Precision**	**%** **Accuracy**		**Mean ^a^ ± SD**	**Precision**	**%** **Accuracy**		**Mean ^a^ ± SD**	**Precision**	**%** **Accuracy**		**Mean ^a^ ± SD**	**Precision**	**%** **Accuracy**
40	39.94 ± 0.72	1.80	99.85	60	59.61 ± 1.06	1.78	99.34	60	59.60 ± 1.14	1.91	99.33	60	59.80 ± 0.62	1.03	99.66
200	199.8 ± 0.42	0.21	99.90	200	199.8 ± 0.83	0.42	99.89	200	199.7 ± 0.48	0.24	99.84	200	199.7 ± 0.64	0.32	99.87
400	399.8 ± 0.44	0.11	99.95	400	399.3 ± 0.86	0.22	99.96	400	400.4 ± 0.60	0.15	100.1	400	400.3 ± 0.56	0.14	100.1

^a^ Mean of three determinations.

**Table 4 molecules-27-09038-t004:** Precision and accuracy calculated from stability experiments for VNB.

Parameter	Conc (ng ml^−1^)	Urine			Plasma			HLMs		
		Mean ^a^ ± SD	% Recovery	% Precision	Mean ^a^ ± SD	% Recovery	% Precision	Mean ^a^ ± SD	% Recovery	% Precision
Freeze–thaw stability (3 cycle, −20 °C)	60	59.50 ± 0.87	99.17	1.46	59.32 ± 1.06	98.87	1.03	59.85 ± 1.06	99.76	1.77
	400	399.9 ± 0.51	99.97	0.13	399.9 ± 1.21	99.97	0.17	399.8 ± 0.67	99.94	0.17
Room temp. stability (8 h at RT)	60	60.13 ± 1.03	100.2	1.71	59.67 ± 1.15	99.44	1.94	59.95 ± 0.08	99.91	0.14
	400	399.9 ± 1.01	99.98	0.25	399.6 ± 0.51	99.89	0.13	399.9 ± 1.02	99.98	0.26
Stored for 24 h at 4 °C (24 h at 4 °C)	60	60.33 ± 1.15	100.6	1.91	60.17 ± 1.01	100.3	1.68	60.32 ± 1.17	100.5	1.94
	400	400.7 ± 0.96	100.2	0.24	399.8 ± 0.96	99.94	0.24	400.7 ± 1.04	100.2	0.26
Long-term stability (30 days at −20 °C)	60	59.92 ± 1.04	99.87	1.74	60.81 ± 0.91	101.3	1.49	60.22 ± 0.77	100.4	1.28
	400	399.2 ± 1.03	99.81	0.26	399.7 ± 0.60	99.93	0.15	400.5 ± 0.64	100.1	0.16

^a^ Mean of three determinations.

**Table 5 molecules-27-09038-t005:** Determination of VNB in pharmaceutical dosage forms.

Pharmaceutical Dosage Forms	Caprelsa (Vandetanib)^®^ (300 mg) Tablets			Reference Method [8]
Parameters	Taken (ng mL^−1^)	Found ^a^	% Recovery	% Recovery
	40	39.80	99.50	98.00
	100	100.0	100.0	100.0
	400	399.0	99.75	99.33
Mean ^a^			99.75	99.11
±S.D.			0.250	1.018
*n*			3.000	3.000
*t*-test			0.71 (2.13) ^b^	
F-ratio			16.3 (19.0) ^b^	

^a^ Average of three determinations. ^b^ The values in parenthesis are the corresponding theoretical values of t and F at *p* = 0.05.

## Data Availability

All experimental data were generated in-house, and no paper mill was used. All data are available within the manuscript.

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
