# Peer review of "A Novel Green Micellar HPLC-UV Method for the Estimation of Vandetanib in Pure Form, Human Urine, Human Plasma and Human Liver Microsomes Matrices with Application to Metabolic Stability Evaluation"

_molecules, 2022, doi:10.3390/molecules27249038_

Round 1

Reviewer 1 Report

- Whether the Authors did not need the consent of the bioethics committee (except patients consent) to study human serum and urine. The majority of studies on humans (for example on the hair or feces samples or even the conducting a survey) require such consent. Whether the study was performed in accordance with the Declaration of Helsinki. Information about it should be added into Material and methods

- I know that this is a methodology article, but whether there were not enough samples (3 urine and serum samples) to draw appropriate conclusions?

- The article would be more readable if the results and discussion were presented as separate chapters

- What is the clinical significance of the study. This significance should be underlined in conclusion.

- Previous investigations on the vandetanib levels in human organism should be discussed  in more detail

Author Response

Reviewer 1

Dear reviewer,

Thank you very much for your time and your comments. Please find the replies to your valuable comments hoping that it would be good as you requested.

Best regards

- Whether the Authors did not need the consent of the bioethics committee (except patient’s consent) to study human serum and urine. The majority of studies on humans (for example on the hair or feces samples or even the conducting a survey) require such consent. Whether the study was performed in accordance with the Declaration of Helsinki. Information about it should be added into Material and methods.

Reply:

The study design used in vitro experiments with commercially available from Sigma company human liver microsomes exempts it from the need of ethical approval. For plasma and urine, from healthy volunteers were supplied by a local Hospital, after consent of volunteers and the study was conducted according to the guidelines of the Declaration of Helsinki.

- I know that this is a methodology article, but whether there were not enough samples (3 urine and serum samples) to draw appropriate conclusions?

Reply:

This is an in vitro experiments, we used three urine and serum samples to confirm there is no effect of the matrix on the drug analysis. All experiments were performed by spiking the matrix (plasma and urine or HLMs), so we used three batches to confirm there in difference on results if we used different sources of biological fluids but we used 12 time of injection or 6 times to confirm results.

- The article would be more readable if the results and discussion were presented as separate chapters

Reply:

Discussion was separated as requested.

- What is the clinical significance of the study. This significance should be underlined in conclusion.

Reply:

The requested in formation was updated:

The low CLint (11.0 µL/min/mg) and the long in vitro t ½ (63.01 min) of VNB in HLMs revealed that VNB is slowly excreted from the blood by the liver that lead to side effects that matched with the literature. So the VNB level in human plasma and urine should be monitored using the developed HPLC-UV analytical methodology to avoid drug accumulation leading to side effects as was proposed from the metabolic stability study. No significant interference from matrix (urine, plasma and HLMs) components were observed. The suggested methodology revealed the benefits of being a green, simple, reliable and economic. The established HPLC-UV analytical methodology could be applied also for the quantification of VNB in human plasma and human urine, so it could be used in pharmacokinetic and therapeutic drug monitoring studies in humans.

- Previous investigations on the vandetanib levels in human organism should be discussed in more detail

Reply:

Requested information were updated in the revised version of the manuscript.

There are limited number of reported methods for the quantification of VNB as spectrofluorimetry and LC/MS [7, 8]. Both published analytical methodology were used to estimate VNB in rat liver microsomes, plasma and urine using fluorescence detector or LC-MS/MS detectors. Also, VNB was estimated either alone or in combination with other drugs [9-13] Micellar liquid chromatographic analytical methodology is one mode of eco-friendly analytical chemistry and reported for separation of number of drugs [14-16]. The VNB pharmacokinetic studies in humans were performed in phase I dose escalation studies [17] and in phase II trials at VNB doses (100 and 300 mg/day). [18] The outcomes of these reported studies revealed that VNB is widely distributed, slowly absorbed, and is slowly excreted with half-life (t1/2) of 120 hours [19].

Reviewer 2 Report

In this article, the authors perfectly introduced an Eco-friendly method employing HPLC-UV to detect Vandetanib in various matrices. The design principle and the innovation of how to accomplish the detection concept were depicted in great detail, which is logical, robust, and elaborate. The validation process is comprehensive. All the conclusions were derived from corresponding experiment results.

The authors developed and validated a Vandetanib detection method employing UPLC-UV method. This topic is not original but practical and relevant in the field. It addressed a gap in the field since few research about using UPLC-UV. All the conclusions are consistent with the evidence. The logic is smooth, and the evidence is powerful. Every reference is appropriate. 

Therefore, I recommend accepting this article after a minor revision.

1.     In section 2.4, “ACN” was used for the first time. However, the full version appeared in section 2.8. Please correct it.   

2.     Please correct the subtitle of 3.3 

Thanks for your wonderful work. Warm regards.

Author Response

Reviewer 2

Dear reviewer,

Thank you very much for your time and your comments. Please find the replies to your valuable comments hoping that it would be good as you requested.

Best regards

In this article, the authors perfectly introduced an Eco-friendly method employing HPLC-UV to detect Vandetanib in various matrices. The design principle and the innovation of how to accomplish the detection concept were depicted in great detail, which is logical, robust, and elaborate. The validation process is comprehensive. All the conclusions were derived from corresponding experiment results.

The authors developed and validated a Vandetanib detection method employing UPLC-UV method. This topic is not original but practical and relevant in the field. It addressed a gap in the field since few research about using UPLC-UV. All the conclusions are consistent with the evidence. The logic is smooth, and the evidence is powerful. Every reference is appropriate.

 Dear reviewer,

Thank you very much for your comments. All

Therefore, I recommend accepting this article after a minor revision.

  1. In section 2.4, “ACN” was used for the first time. However, the full version appeared in section 2.8. Please correct it.

Reply: It was corrected as requested and it was updated in the revised manuscript.

  1. Please correct the subtitle of 3.3

Reply: It was corrected as requested and it was updated in the revised manuscript.